# The association between a rotator cuff tendon tear and a tear of the long head of the biceps tendon: Chart review study

**Abdulrahman Alraddadi**[1,2]*, **Bader Aldebasi**[2], **Bander Alnufaie**[1], **Mohammed Almuhanna**[1], **Mohammed Alkhalifah**[1], **Motaz Aleidan**[1], **Yousef Murad**[1], **Awad M. Almuklass**[1,2], **Altayeb A. Ahmed**[1,2]

**1** Department of Basic Medical Sciences, College of Medicine, King Saud bin Abdulaziz University for Health Sciences, Riyadh, Saudi Arabia, **2** King Abdullah International Medical Research Center (KAIMRC), Ministry of National Guard – Health Affairs, Riyadh, Saudi Arabia

* aalraddadi.2012@gmail.com, raddadia@ksau-hs.edu.sa

**Data Availability Statement:** All data generated or analyzed in this study are available at https://doi.org/10.7910/DVN/F3VD46.

## Abstract

Rotator cuff (RC) and long head of the biceps tendon (LHBT) tears are common shoulder problems presented to the orthopedic clinic. The aim of this study was to assess the association between RC and LHBT tears among a Saudi population sample. A total of 243 patients who were diagnosed with shoulder pain due to RC or LHBT tear between 2016 and 2018 using a magnetic resonance imaging scan were included in this study. Females comprised 66% of the sample, and 59% (n = 143) of the shoulders were on the right side. The mean age of the patients was 58 ± 11 years, ranging from 23 to 88 years. A significant association was detected between the LHBT and RC tears (P < 0.001). Out of 26 cases showing RC and LHBT tears, 81% had a full thickness tear, whereas 19% had a partial tear. The LHBT tears were presented significantly in 48% of cases with at least two completely torn RC compared to 10% in cases with one completely torn RC (P < 0.001). The LHBT tear was significantly observed in shoulders with RC tears including the tendons of subscapularis, supraspinatus, and infraspinatus, but not the teres minor (P < 0.001). Both types of tears were presented significantly in senior patients aged more than 65 years compared to younger patients (P < 0.01). Thus, the LHBT should be assessed carefully in shoulders with more than one RC tear or in chronic cases.

## Introduction

Glenohumeral joint injuries are common musculoskeletal disorders. They are ranked as the third most common presentation to orthopedic clinics following disorders of the back and neck [1]. These injuries may involve bones, muscles, or ligaments. This can be attributed to the fact that the shoulder joint is very mobile but has minimal congruity between its articular surfaces [2]. The rotator cuff (RC) muscles play a pivotal role in this joint's stability by ensuring proper orientation of the head to the fossa, especially in abduction, in addition to their role in humeral movements (flexion, extension, rotation, and adduction) [3, 4].

**Funding:** The author(s) received no specific funding for this work.

**Competing interests:** The authors have declared that no competing interests exist.

However, the long head of the biceps tendon (LHBT) brachii contribution to the joint's stability is not fully understood [4]. Anatomically, the LHBT originates proximally from the scapular supraglenoid tubercle and superior labrum [2]. It is formed of wide and flat intraarticular and small and rounded extraarticular parts. In neutral and medial rotation of the shoulder, the LHBT runs through the capsule, initially curving over the anterosuperior part of the head of the humerus before passing in hiatus between the subscapularis and supraspinatus tendons accompanied by the coracohumeral ligament [2]. The LHBT leaves the sulcus intertubercularis at the level humeral neck to join the short head distally. In this course, the LHBT provides extra support to the RC anterosuperior part and aids in maintaining glenohumeral normal relationship [5].

A RC tear is an incapacitating clinical condition and assumes a pivotal part in deciding health status as per the 36-item Short Form (SF-36) survey [6]. Studies show that RC tendon injuries can be isolated or involve multiple tendons, and these tears can be partial or complete. Among the most common causes of a RC tear is subacromial impingement syndrome [7]. The supraspinatus tendon is the most common RC tear and presents as an isolated condition in about half of patients. Due to their proximity to RC muscles, RC tears are associated with LHBT tears and subluxation [8]. This is more established for the tears involving the subscapular tendon despite it being unusual [9]. In contrast, anterior and superior RC tears associated with LHBT tears are less established and are usually highlighted as massive tears when including two or more tendons or when the RC tear exceeds 5 cm in its maximum diameter [10].

Various studies have assessed the connection between RC and LBHT tears. However, the sample size of most of these studies is small, and no such study has been conducted on Saudi Arabian Patient. Therefore, the aim of the study is to assess the prevalence and spectrum of RC tears and their association with LHBT tears in patients presenting with shoulder pain due to these conditions, in addition to depicting various patterns of pathologies that may help in improving radiological interpretation accuracy and managing patients presenting with shoulder pain.

## Materials and methods

### Study design

A chart review for RC and bicep brachii tendon findings during shoulder pain clinical assessment and magnetic resonance imaging (MRI) investigation formed the basis of this retrospective analysis. The study was conducted in accordance with the Declaration of Helsinki, and approved by the Institutional Review Board of King Abdullah International Medical Research Center, the study number: SP18/314/R and IRB approval date in 31 July 2018. Following IRB approval from King Abdullah International Medical Research Center (KAIMRC), a search was performed on the electronic health records of patients at King Abdulaziz Medical City, National Guard-Health Affairs, Riyadh, Saudi Arabia from 2016 to 2018.

### Study subjects

Initially, over 600 patient records were reviewed with acceded date between 1st September 2018 to 31st August 2019 through the Research Data Management Office at KAIMRC. Those who were diagnosed as a case of partial or complete RC tear and/or LHBT tear were included. Exclusion criteria included patients with RC or LHBT tears following motor vehicle accidents, previous surgery, or tumors.

Of those patients initially reviewed, only 243 patients were eligible. The variables collected included the RC tear type (partial or complete), the number of RC injured muscles, LHBT tear, cases with bursal inflammation or damage, glenoid labrum tear, osteoarthritic changes,

presence of acromial changes, evidence of degenerative process in the greater tuberosity, and topographical data such as age, gender, and the side of the affected limb. The age of 65 years was selected as a demarking age between the two groups based on a previous study showing that this age is associated with higher incidence [11].

### Study setting

Utilizing standard intermediate-weighted, fat-suppressed, and nonfat-suppressed imaging sequences in the axial, coronal, and sagittal planes with 0.5 to 3 mm slices, MRI was carried out on 1.5 T and 3 T scanners. The scapular body, which was recognized on the scout images, was obtained parallel and perpendicular to the coronal and sagittal plane images, respectively. All MRI tests carried out were not arthrograms using MR imaging. Every MRI report included information about the RC muscles, long head of biceps, bursa inflammation, glenoid labrum condition, acromioclavicular osteoarthrosis, acromial spur, and greater tuberosity degeneration. All reports were written by expert radiologists. This information, in addition to the demographic data, was taken by medical students and entered into an Excel (Microsoft 365) spreadsheet.

### Statistical analysis

Descriptive and analytical statistics were performed for the data collected by IBM Statistical Package for the Social Sciences (SPSS) version 25. The chi-square test was used to assess the association between the RC and LHBT tears in relation to number and severity. Furthermore, the association between the LHBT and RC tears according to the torn muscles was assessed. Association of these variables and the independent variables such as age, gender, and side affected was also evaluated. An odds ratio (OR) and 95% confidence interval (CI) were used to measure the risk of developing LHBT tears using logistic regression tests. The effect of age on tear incidence was assessed using an independent t-test, which measured the difference in the age of patients based on the incidence of tear. The statistical analysis results were considered significant if the probability (P) value was $< 0.05$.

## Results

### The prevenance of the RC and LHBT tears according to age, sex, and side

Of the 243 eligible patients, 160 (66%) were female and 83 (34%) were male. Right shoulders were involved in 143 cases (59%) and left shoulders affected in 100 (41%) of cases. The average age of female patients was 58 years (range 23–86 years), while the average for males was 59 years (range 31–88 years). Most of the subjects (71%) were $< 65$ years (Table 1).

The RC partial tears were found to be more common than complete tears (54% compared to 46%). One tendon tear was the most common pattern of RC partial and full-thickness tears, seen in 77% and 77.7% of patients, respectively. The full thickness RC tears were most prevalent in the supraspinatus muscle (112 cases). RC tears were significantly ($P = 0.002$) associated with the age of the subjects. The partial thickness RC tears were observed to be more common in patients $< 65$ years, while full thickness tears were more common in patients $\geq 65$ years.

The LHBT tears were evident in 26 (11%) cases. No significant correlation existed between the LHBT tears and gender or shoulder side ($P > 0.050$). Out of these 26 cases, the LHBT tears were detected in 62% of females compared to 38% of males, as well as in 58% of right-sided shoulders versus 42% of left-sided shoulders. The LHBT tears were found to be significantly ($P < 0.001$) more common in the senior age group (62%) compared to the younger age group (38%).

**Table 1. Descriptive collected data according to age, sex, and shoulder side.**

| Findings/Variables | N = 243 (%) | Age | | Sex | | Side | |
|---|---|---|---|---|---|---|---|
| | | < 65 Ys n = 173 (71%) | ≥ 65 Ys n = 70 (29%) | Male n = 83 (34%) | Female n = 160 (66%) | Right n = 143 (59%) | Left n = 100 (41%) |
| **RC Partial tears**: | 131 (54) | 104 (79)* | 27 (21) | 44 (34) | 87 (66) | 73 (56) | 58 (44) |
| One tendon | 101 (77) | 80 (79) | 21 (21) | 37 (37) | 64 (63) | 58 (57) | 43 (43) |
| Two tendons | 24 (18) | 20 (83) | 4 (17) | 6 (25) | 18 (75) | 11 (46) | 13 (54) |
| Three tendons | 6 (5) | 4 (67) | 2 (33) | 1 (17) | 5 (83) | 4 (67) | 2 (33) |
| **RC Full tears**: | 112 (46) | 69 (62) | 43 (38)* | 39 (35) | 73 (65) | 70 (62.5) | 42 (37.5) |
| One tendon | 87 (77.7) | 60 (69) | 27 (31) | 27 (31) | 60 (69) | 51 (59) | 36 (41) |
| Two tendons | 22 (20) | 7 (32) | 15 (68) | 10 (45) | 12 (36) | 17 (77) | 5 (23) |
| Three tendons | 3 (2.7) | 2 (67) | 1 (33) | 2 (67) | 1 (33) | 2 (67) | 1 (33) |
| Subscapularis | 6 (5) | 2 (33) | 4 (67) | 3 (50) | 3 (50) | 3 (50) | 3 (50) |
| Supraspinatus | 112 (100) | 69 (62) | 43 (38) | 39 (35) | 73 (65) | 70 (62.5) | 42 (37.5) |
| Infraspinatus | 22 (20) | 9 (41) | 13 (59) | 11 (50) | 11 (50) | 18 (82) | 4 (18) |
| Teres Minor | 0 | 0 | 0 | 0 | 0 | 0 | 0 |
| LHBT Tears | 26 (11) | 10 (38) | 16 (62)* | 10 (38) | 16 (62) | 15 (58) | 11 (42) |
| Bursa Degeneration | 173 (71) | 124 (72) | 49 (28) | 57 (33) | 116 (67) | 103 (60) | 70 (40) |
| Glenoid Labrum Tears | 32 (13) | 20 (62.5) | 12 (37.5) | 14 (44) | 18 (56) | 18 (56) | 14 (44) |
| Acromioclavicular Joint Osteophyte | 201 (83) | 141 (70) | 60 (30) | 66 (33) | 135 (67) | 128 (64)* | 73 (36) |
| Acromial Spur | 78 (32) | 63 (81)* | 15 (19) | 27 (35) | 51 (65) | 49 (63) | 29 (37) |
| Greater Tuberosity Degeneration | 47 (19) | 35 (74) | 12 (26) | 19 (40) | 28 (60) | 29 (62) | 18 (38) |

* p value < 0.05

Subacromial bursal inflammation or damage was observed in 173 (71%) cases with no significant association with age, sex, or side. The glenoid fibrocartilaginous tear was found in 13% of cases. Acromioclavicular joint osteoarthritic changes were recognized in most cases (83%) and were significantly (P < 0.001) associated with right-side (64%) cases compared to left-side (36%) cases. Acromial spur was found in 32% of cases and was significantly (P = 0.023) more prevalent in younger aged (81%) cases compared to (19%) cases in patients greater than 65 years old. The greater tuberosity degeneration was evident in 19% of cases.

### The relationship between the RC and LHBT tears according to tear type. Number of torn muscles, and the torn muscles

As depicted in Table 2, LHBT tears were significantly prevalent in shoulders with complete tears (19%) compared to shoulders with partial tears (4%), (P < 0.001, OR = 5.82, 95% CI = 2.11–16.0). Furthermore, there were substantially more LHBT tears (38%) in shoulders with three RC tears compared to shoulders with two (13%) or one (5%); P 0.001, OR = 4.19 and 11.53, 95% CI = 1.41–12.48 and 4.10–32.42), respectively. LHBT tears were significantly more common in shoulders with at least two complete RC tears than in shoulders with one full RC tear (P 0.001, OR = 8, 95% CI = 2.81–22.74). The tendons of the subscapularis, supraspinatus, and infraspinatus were all completely torn in only three patient shoulders, whereas 100% of patient shoulders had LHBT tears.

The association between the LHBT and RC tears is presented in Table 3. The results showed significant numbers of LHBT tears in shoulders with RC tears, including the tendons of subscapularis, supraspinatus, and infraspinatus, but not teres minor (P < 0.001). LHBT tears were

**Table 2. The association between the long head of biceps tear and rotator cuff tears according to rotator cuff tear type and number of torn muscles.**

| Independent Variable | LHBT | | Comparison | P Value | Odds Ratio (%) | 95% C.I. for OR |
|---|---|---|---|---|---|---|
| | No Tear (n = 217) | Tear (n = 26) | | | | |
| **Tear type (%)** | | | | | | |
| Partial (n = 131) | 126 (96) | 5 (4) | | < 0.001 | 5.82 | 2.11–16.0 |
| Full thickness (n = 112) | 91 (81) | 21(19) | | | | |
| **Number of torn muscles: including partial & full-thickness tears (%)** | | | | | | |
| One (n = 159) | 151 (95) | 8 (5) | a vs. b | 0.115 | 2.75 | 0.95–7.99 |
| Two (n = 55) | 48 (87) | 7 (13) | a vs. c | < 0.001 | 11.53 | 4.10–32.42 |
| Three (n = 29) | 18 (62) | 11 (38) | b vs. c | 0.007 | 4.19 | 1.41–12.48 |
| **Number of tendons with full thickness cuff tears (%)** | | | | | | |
| One (n = 87) | 78 (90) | 9 (10) | | < 0.001 | 8 | 2.81–22.74 |
| At least Two (n = 25) | 13 (52) | 12 (48) | | | | |

**Table 3. The association between the long head of bices tears and rotator cuff tears according to the torn muscles.**

| Independent Variable | LHBT | | Comparison | P Value | Odds Ratio (%) | 95% C.I. for OR |
|---|---|---|---|---|---|---|
| | No Tear (n = 217) | Tear (n = 26) | | | | |
| Subscapularis | | | | | | |
| a. No tear (n = 184) | 175 (95) | 9 (5) | a vs. b | < 0.001 | 6.14 | 2.53–15.80 |
| b. Partial (n = 53) | 40 (75) | 13 (25) | a vs. c | < 0.001 | 38.89 | 6.27–241.1 |
| c. Full thickness (n = 6) | 2 (33) | 4 (67) | b vs. c | 0.103 | 6.15 | 1.01–37.56 |
| Supraspinatus | | | | | | |
| a. No tear (n = 11) | 9 (82) | 2 (18) | a vs. b | 0.112 | 0.11 | 0.02–0.78 |
| b. Partial (n = 120) | 117 (98) | 3 (2) | a vs. c | 0.987 | 1.04 | 0.21–5.16 |
| c. Full thickness (n = 112) | 91 (81) | 21 (19) | b vs. c | < 0.001 | 9 | 2.60–31.11 |
| Infraspinatus | | | | | | |
| a. No tear (n = 180) | 168 (93) | 12 (7) | a vs. b | 0.846 | 1.11 | 0.30–4.11 |
| b. Partial (n = 41) | 38 (93) | 3 (7) | a vs. c | < 0.001 | 14 | 5.05–38.84 |
| c. Full thickness (n = 22) | 11 (50) | 11 (50) | b vs. c | < 0.001 | 12.67 | 2.99–53.58 |
| Teres Minor | | | | | | |
| a. No tear (n = 241) | 215 (89) | 26 (11) | | 0.749 | 2.07 | 0.09–47.07 |
| b. Partial (n = 2) | 2 (100) | 0 (0) | | | | |

found in more shoulders with subscapularis partial tears than those with no tendon tears and/ or full thickness tears. Shoulders that had full thickness tears in the supraspinatus showed a significant number of LHBT tears (81%) compared to those with partial tears, but not to supraspinatus with no tears. In addition, shoulders with full thickness tears in the infraspinatus had significant numbers of LHBT tears (42%) compared to those with partial tears (12%) and no tears (46%).

Furthermore, Table 4 presents the association between LHBT tears and RC full thickness tears according to the presentation of tears in the study samples. A single RC full thickness tear was seen only in the supraspinatus (n = 87 (36%)). The LHBT tears were detected in 10% of these shoulders. More LHBT tears were also seen in shoulders with double RC tendons and full thickness tears: supraspinatus and infraspinatus (42%), and supraspinatus and subscapularis (33%). Only three patient shoulders showed multiple RC tendon full thickness tears,

**Table 4. The association between the long head of biceps tear and rotator cuff full-thickness tears according to presentation of tears among study subjects.**

| Independent Variable | LHBT | | P Value | Odds Ratio (%) | 95% C.I. for OR |
|---|---|---|---|---|---|
| | No Tear (n = 217) | Tear (n = 26) | | | |
| **Single rotator cuff tendon tears[a]** | | | | | |
| Supraspinatus (n = 87) | 78 (90) | 9 (10) | | | |
| **Multiple rotator cuff tendon tears[a]** | | | | | |
| Supra. + Infra (n = 19) | 11 (58) | 8 (42) | | | |
| Supra. + Subs. (n = 3) | 2 (67) | 1 (33) | | | |
| Supra. + Infra. + Subs. (n = 3) | 0 | 3 (100) | | | |
| **With & without subscapularis tears (%)** | | | | | |
| With Subs. Tears (n = 6) | 2 (33) | 4 (67) | 0.012 | 0.10 | 0.02–0.56 |
| Without Subs. Tears (n = 106) | 89 (84) | 17 (16) | | | |
| **With & without Infraspinatus tears (%)** | | | | | |
| With Infra. Tears (n = 22) | 11 (50) | 11 (50) | < 0.001 | 0.13 | 0.04–0.36 |
| Without Infra. Tears (n = 90) | 80 (89) | 10 (11) | | | |

[a]: The number of specimens in the first and second rows cells were not meet Cochran's criteria for accepting the results of Chi Square test.

including supraspinatus, subscapularis, and infraspinatus. Each of these three shoulders also had LHBT tears. A significant association was detected between the LHBT tear and subscapularis full thickness tear (P = 0.012, OR = 0.10, 95% CI = 0.02–0.56). Shoulders with subscapularis full thickness tears had more LHBT tears than shoulders without subscapularis tears, 67% compared to 16%, respectively. A significant association also was identified between the LHBT tears and infraspinatus full thickness tears (P < 0.001, OR = 0.13, 95% CI = 0.04–0.36). Shoulders with infraspinatus full thickness tears showed more LHBT tears than shoulders without infraspinatus tears, 50% compared to 11%, respectively.

## The relationship between the RC and LHBT tears and changes seen in subacromial bursa and acromioclavicular joint

RC tears and LHBT tears were more frequently observed in those patients presenting with subacromial damage than those who only had inflammation (Table 5). Moreover, subacromial damage showed similar prevalence in cases of partial and full thickness RC tears.

**Table 5. The association between each of RC and LHBT tears and degeneration of subacromial bursa and acromioclavicular joint.**

| Independent Variable | RC | | | | | LHBT | | | | |
|---|---|---|---|---|---|---|---|---|---|---|
| | Partial Tear (n = 131) | Full Tear (n = 112) | P Value | Odds Ratio (%) | 95% C.I. for OR | No Tear (n = 217) | Tear (n = 26) | P Value | Odds Ratio (%) | 95% C.I. for OR |
| **Subacromial Bursa (%)** | | | | | | | | | | |
| No Damage (n = 70) | 40 (31) | 30 (27) | 0.525 | 1.20 | 0.69–2.10 | 62 (29) | 8 (31) | 0.803 | 0.9 | 0.37–2.18 |
| Damage (n = 173) | 91 (69) | 82 (73) | | | | 155 (71) | 18 (69) | | | |
| **Acromioclavicular Joint (%)** | | | | | | | | | | |
| No Changes (n = 42) | 30 (23) | 12 (11) | 0.012 | 2.48 | 1.2–5.12 | 40 (18) | 2 (8) | 0.172 | 2.71 | 0.62–11.94 |
| Ost. Changes (n = 201) | 101 (77) | 100 (89) | | | | 177 (82) | 24 (92) | | | |

**Table 6. Age difference between the patients with and without long head of biceps tears.**

| Variables | Mean Age (SDV) | P Value |
|---|---|---|
| Patients with RC Tear (n = 217) | 57.27 ± 10.67 | <0.001 |
| Patients with RC & LHBT Tear (n = 26) | 68.23 ± 10.68 | |

Acromioclavicular joint osteoarthritic changes were significantly (P < 0.012) higher in full thickness tears compared to those of partial tears of RC tears. Furthermore, they are more common in cases of LHBT tears (92%) compared to no LHBTs cases (82%).

## The patients age difference according to the presence of LHBT tear

The prevalence of RC and LHBT tears varied significantly depending on the age of the patients (Table 6). Patients with RC and LHBT tears were significantly older than those had only RC tears, with mean ages of 68.23 ± 10.68 compared to 57.27 ± 10.67 (P < 0.001), respectively.

## Discussion

The findings of this study among Saudi subjects showed a strong association between the RC and LHBT tears. The number of RC tendons involved and the severity of the tear were found to both strongly impact LHBT, with the number of RC muscles being directly proportionate to the prevalence of LHBT tears. The subscapularis muscle tendon tear was the most common to be associated with LHBT tear.

The current study showed that overall, partial RC tears are more common than full thickness tears. While full RC tears were more common in patients of an older age, the partial RC tears mostly affected patients in their active physical life (< 65 years). We suggest that these symptoms are felt less in patients of a younger age, but this diminishes with time. This concurs with previous cadaveric and radiological studies, which have shown that increasing age is positively associated with full thickness tears [12, 13]. The degenerative effect of age accompanied by chronic microtrauma has been suggested as one of the mechanisms of developing partial tears, which will be converted to a full thickness tear by tensile retraction of intact muscle fibers. Apoptotic and remodeling tissue changes following chronic inflammatory processes and decreased blood supply can also be responsible for such an effect [13]. However, these results should be considered with caution as our sample was extracted from symptomatic patients presenting to a hospital seeking medical advice, hence, many asymptomatic patients may not be represented. A previous report suggested that only half of patients having symptomatic RC tear pathologies will seek advice [14]. Furthermore, even though more severe symptomatology is likely to be encountered in larger tears, previous studies did not show a positive correlation between pain threshold and the size of the tear among affected subjects as pain was evident only in one-third of total thickness [15, 16].

In the present sample, the right shoulder was more frequently involved than the left in both sexes. This can be attributed the fact that hand dominance increases cumulative loading, resulting in higher susceptibility to shoulder pathologies. This finding concurs with previous research showing both symptom prevalence and pathologies were greater in the dominant arm [14, 16]. Furthermore, the findings of the current study showed that most of the affected subjects are females. This can be attributed to the high likelihood of injury in the postmenopausal period [17] and that females have subjectively higher pain levels on having RC tears [18]. However, this finding is confounding; studies have shown that both sexes have equal chances of having RC tears [19].

The most common pathologies encountered with RC tears were acromioclavicular joint osteoarthritic change and subacromial bursal inflammation. These two conditions are chronic in nature and can result in progressive weaking of the RC tendons, which eventually rupture. Older age was significantly associated with LHBT tears. In contrast, all other pathologies were commonly observed in patients of a younger age, with acromial spur being significantly more common. Nontraumatic RC tears have postulated to be a sequel of the repetitive back and forth movement of the tendons due to osteoarthritic changes in the coracoacromial joint, abnormalities in coracoacromial arch appearance, or developmental anomalies such as Os acromiale [20]. Furthermore, coracoacromial spurs, abnormal coracoid morphology, increased pillar angulation, decreased interpillar distance, or shortened pillar length can contribute to RC tears [21]. In addition, inflammation involving shoulder bursae can result in adhesions or impingement of RC tendons [21].

This study confirmed that there is an association between RC and LHBT tears. This is in accordance with previous research that has demonstrated an epidemiological association between RC and LHBT tears that ranged between 21% and 45% [22–26]. Various pathologies were observed in these cases, in addition to bicep tendonitis, instability, and pulley lesions, and RC tear-affected shoulders exhibited biceps tendonitis [22, 24, 27, 28]. In accordance with prior studies, the present study confirms the close link between the two muscles, albeit with a lower prevalence (11%). The type of RC tear has a substantial effect on the occurrence of LHBT tears where shoulders with a complete RC tear were more likely to have a LHBT tear than shoulders with a partial RC tear (P < 0.001, OR = 5.82, 95% CI = 2.11–16.0). This finding agrees with previous studies indicating a direct positive association between intensity of the RC tears and LHBT involvement [23, 26]. Additionally, bicep tendon pathology was detected in all chronic cases diagnosed with RC tears for more than three months. Other studies found a correlation between the size of the RC tear, the extent of instability of the LHBT, and its tendon low mechanical properties [25, 28, 29]. Prolonged shoulder impingement would result in tendinitis and/or a partial tendinous rupture, which will proceed to a full tear affecting further subacromial tissue, including shoulder bursae, RC tendons, LHBT, and the glenohumeral ligaments [30].

With respect to the number of torn RC muscles, LHBT tears are more likely to occur in shoulders with more than one torn RC muscle. These findings are consistent with literature reporting a higher incidence of LHBT tears in shoulders with multiple RC tendon tears, rather than a single tendon tear. Concurring with previous literature, in this study, the three patients who had a combined full thickness RC tear of subscapularis, supraspinatus, and infraspinatus also showed LHBT tears [23, 29]. Furthermore, we found that shoulders with full thickness tears involving the supraspinatus and infraspinatus tendons had a higher incidence of LHBT tears than those with full thickness RC tears involving the supraspinatus and subscapularis tendons. This indicates that tears of the supraspinatus and infraspinatus tendons had a greater effect on the mechanical properties of the bicep tendons than tears of the subscapularis and supraspinatus tendons. Our finding contradicts a previous study, which showed that tears in tendons of the supraspinatus and subscapularis are more likely to be associated with severe LHBT pathology than those of the supraspinatus and infraspinatus [23]. As multiple full thickness tears are more common in chronic cases, LHBT tears should be assessed carefully both clinically and radiologically when surgeons encounter more than one RC tear or in chronic cases.

The findings of this study showed that the supraspinatus muscle is the most common muscle involved in partial and full thickness RC tears and that a LHBT tear is more associated with supraspinatus full thickness tear and subscapular partial tears. LHBT and supraspinatus tears were closely associated with a reported prevalence ranging from 22% to 78.5% in one systemic

review [31]. Another systematic review indicated that the supraspinatus was involved in 84% of cases, and this was attributed to the supraspinous critical position and the amount of pressure usually applied to it [32]. Another study that assessed the association between LHBT and RC tears reported that 85% of patients with LHBT tears had some degree of supraspinatus or subscapularis tendon tears [33]. In our study, there was no association between LHBT tears and teres minor tendon tears, which agrees with previous findings [22]. On the other hand, the study reported that LHBT tears are associated with infraspinous tears in 50%, subscapularis in 28%, and supraspinatus in 22% of cases [22]. Our findings may be due to the close anatomic proximity between the supraspinatus and subscapularis muscles, with the LHBT running in the rotator interval between the tendons of the subscapularis anteriorly and the supraspinatus posteriorly, which place the long biceps tendon in the impingement zone beside the RC tendons [3].

It is evident that older people are more prone to LHBT tears with RC tears. In the present study, a significant association was identified between age and RC and LHBT tears. Due to the degenerative process caused by aging and the other pathological conditions that can damage shoulder joints, LHBT tears were more pronounced in the elderly population. As previously established, the anatomical link between the shoulder joint's components plays a crucial part in shoulder disease, as it is the origin and insertion point for many of the muscles. A torn RC muscle can result in the creation of scar tissue through the healing process, which anatomically narrows the pathway for the long head of the biceps, causing tendinopathy and tear; elderly patients will be more susceptible to these changes [13].

According to our knowledge, this is the first study undertaken in the Kingdom of Saudi Arabia. Due to its ability to view soft tissues, MRI was chosen as the radiological modality of choice to explore the presence of tears in the current study. Nevertheless, using MRI as a diagnostic tool forms one of the limitations of this study as MRI may not be performed on all patients presenting with shoulder discomfort, hence limiting the sample size for evaluating RC tears. A further advantage of this study was the removal of individuals with significant injuries, as the objective was to monitor the neutral degenerative process in the absence of physical harm. Only radiological results documented by radiologists and confirmed by orthopedic physicians served as the basis for data collection. As various forms of injuries are associated with individual jobs or lifestyles, a clear perspective can be attained if the study incorporates patients' medical data and lifestyles. The most important finding of this study was that it confirmed the relationship between RC and LHBT tears, as reported in the previous studies. However, the relationships and associated risk factors must be validated. Therefore, further wide scale studies are recommended to examine the risk factors and causes of RC and LHBT tears, to comprehend the pathophysiology and development of muscle tears, and to enhance the therapeutic management of these cases to maximize the care outcome for patients.

## Conclusion

The current study investigated the association between RC and LHBT tears through reviewing the MRI findings of patients attending the orthopedic clinic due to shoulder pain. The results indicate that 243 of the patients had RC tears and 26 had LHBT tears. A significant association was identified between RC tears and LHBT tears, especially in the elderly group of patients. The prevalence of developing LHBT tears in shoulders with RC tears increased under the influence of the type of RC tear and the number and type of involved RC muscles. The close anatomical relationship between the tendons of the long head of the biceps and the RC, mainly the supraspinous and subscapularis, may explain this association.

## Author Contributions

**Conceptualization:** Abdulrahman Alraddadi, Bader Aldebasi.

**Data curation:** Abdulrahman Alraddadi, Bader Aldebasi, Bander Alnufaie, Mohammed Almuhanna, Mohammed Alkhalifah, Motaz Aleidan, Yousef Murad.

**Formal analysis:** Abdulrahman Alraddadi, Awad M. Almuklass.

**Investigation:** Abdulrahman Alraddadi, Bander Alnufaie, Mohammed Almuhanna, Mohammed Alkhalifah, Motaz Aleidan, Yousef Murad.

**Methodology:** Abdulrahman Alraddadi, Bader Aldebasi, Bander Alnufaie, Mohammed Almuhanna, Mohammed Alkhalifah, Motaz Aleidan, Yousef Murad.

**Project administration:** Bader Aldebasi.

**Supervision:** Abdulrahman Alraddadi, Bader Aldebasi.

**Validation:** Abdulrahman Alraddadi, Altayeb A. Ahmed.

**Writing – original draft:** Abdulrahman Alraddadi, Awad M. Almuklass, Altayeb A. Ahmed.

**Writing – review & editing:** Abdulrahman Alraddadi, Bader Aldebasi, Bander Alnufaie, Mohammed Almuhanna, Mohammed Alkhalifah, Motaz Aleidan, Yousef Murad, Awad M. Almuklass, Altayeb A. Ahmed.

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
