## [Decision Letter · Decision Letter 0]

12 Jan 2024

PONE-D-23-29196The Association Between a Rotator Cuff Tendon Tear and a Tear of the Long Head of the Biceps Tendon: Chart Review StudyPLOS ONE

Dear Dr. Alraddadi,

Thank you for submitting your manuscript to PLOS ONE. After careful consideration, we feel that it has merit but does not fully meet PLOS ONE’s publication criteria as it currently stands. Therefore, we invite you to submit a revised version of the manuscript that addresses the points raised during the review process.

We look forward to receiving your revised manuscript.

Kind regards,

Sabata Martino, Ph.D

Academic Editor

PLOS ONE

Journal Requirements:

Reviewers' comments:

Reviewer's Responses to Questions

**Comments to the Author**

1. Is the manuscript technically sound, and do the data support the conclusions?

Reviewer #1: Yes

Reviewer #2: Yes

2. Has the statistical analysis been performed appropriately and rigorously? 

Reviewer #1: Yes

Reviewer #2: I Don't Know

3. Have the authors made all data underlying the findings in their manuscript fully available?

Reviewer #1: Yes

Reviewer #2: Yes

4. Is the manuscript presented in an intelligible fashion and written in standard English?

Reviewer #1: Yes

Reviewer #2: Yes

5. Review Comments to the Author

Reviewer #1: The manuscript entitled “The Association Between a Rotator Cuff Tendon Tear and a Tear of the Long Head of the Biceps Tendon: Chart Review Study” by Alraddadi et al. addresses an extremely interesting and original topic concerning the association between an injury of the long head of the biceps tendon and the tear of the rotator cuff tendon evaluated on a large group of patients.

In general, the provided text appears clear and concise, presenting key findings from the study. The language is straightforward and effectively communicates the main objective and results.

The abstract clearly presents the study, the introduction is well-written and allows to have a comprehensive vision of the topic. The conclusions are clear and concise.

Comments for the Author:

In my opinion, the work could be further enriched in the materials and methods section. This part could be implemented by dividing the different sections into titled subparagraphs for better reader understanding. The same approach could also be used for the results section.

Please check that table 4 is complete and standardize table 5 by adding the statistic in the table.

Decision: Minor Revision

Reviewer #2: Comments To the authors

. The uniqueness of the current study is called into doubt because the purpose and goals of this investigation have already been made clear in a number of well-published studies as the authors mentioned in the “Discussion” (Lines 224-225) (references number 9,34-37 in the current manuscript). Like the current study, previous studies proved a direct positive association between intensity of the RC tears and LHBT involvement (references number 35 and 37 in the current manuscript). Also, one of the findings in the current study is a

higher incidence of LHBT tears in shoulders with multiple RC tendon tears, rather than a single tendon tear.

Again, this is not anew finding and previously proved in previous studies.

Abstract

. Line 13: “..due to an rotator cuff ..” “an” should be deleted

. Line 20: The meaning of the abbreviation “RC” was not previously mentioned. As a rule, with the abstract considered a spate entity, the abbreviation of any word should be mentioned on its first use, then the abbreviation only should be used afterword.

. The introduction section is lengthy. It needs to be shortened. Kindly focus on three elements of introduction. a. What is known about the topic? (Background)

b. What is not known? (The research problem)

c. Why the study was done? (Justification)

. Lines 113-114: What is the meaning of this sentence: “The presence of one tendon partial tear was the commonest pattern of RC partial and full thickness tears as it

was detected in 77% and 77.7% of patients, respectively.”?

. Line 125: “Acromioclavicular osteoarthritic changes” should be corrected to “Acromioclavicular joint osteoarthritic changes

. Line 139: “tor” is a topographical error should be corrected to “torn”

Line 141: “An examination of the association between the LHBT and RC tears is presented in Table 3.” Should be corrected to “The association between the LHBT and RC tears is presented in Table 3.”

First paragraph of the “Discussion” should summarize the results of the study

“Discussion” is too long and should be shortened.

6. PLOS authors have the option to publish the peer review history of their article (what does this mean?). If published, this will include your full peer review and any attached files.

Reviewer #1: No

Reviewer #2: No

---

## [Author Response · Author response to Decision Letter 0]

2 Feb 2024

Thanks for reviewing the manuscript and for your helpful comments. The editor’s and reviewers’ comments were reviewed and addressed in the Response to Reviewers letter as requested, as well as all changes made in the revised manuscript as requested.

---

## [Editor Report · Decision Letter 1]

26 Feb 2024

The Association Between a Rotator Cuff Tendon Tear and a Tear of the Long Head of the Biceps Tendon: Chart Review Study

PONE-D-23-29196R1

Dear Dr. Alraddadi,

We’re pleased to inform you that your manuscript has been judged scientifically suitable for publication and will be formally accepted for publication once it meets all outstanding technical requirements.

Kind regards,

Sabata Martino, Ph.D

Academic Editor

PLOS ONE

Additional Editor Comments (optional):

no comments
---

## [Editor Report · Acceptance letter]

29 Feb 2024

PONE-D-23-29196R1 

PLOS ONE

Dear Dr. Alraddadi, 

I'm pleased to inform you that your manuscript has been deemed suitable for publication in PLOS ONE. Congratulations! Your manuscript is now being handed over to our production team.

Kind regards, 

on behalf of

Prof. Sabata Martino 

Academic Editor

PLOS ONE